# Heritable capture of heterochromatin dynamics in *Saccharomyces cerevisiae*

**Anne E Dodson[1,2], Jasper Rine[1,2]\***

[1]Department of Molecular and Cell Biology, University of California, Berkeley, Berkeley, United States; [2]California Institute for Quantitative Biosciences, University of California, Berkeley, Berkeley, United States

**Abstract** Heterochromatin exerts a heritable form of eukaryotic gene repression and contributes to chromosome segregation fidelity and genome stability. However, to date there has been no quantitative evaluation of the stability of heterochromatic gene repression. We designed a genetic strategy to capture transient losses of gene silencing in *Saccharomyces* as permanent, heritable changes in genotype and phenotype. This approach revealed rare transcription within heterochromatin that occurred in approximately 1/1000 cell divisions. In concordance with multiple lines of evidence suggesting these events were rare and transient, single-molecule RNA FISH showed that transcription was limited. The ability to monitor fluctuations in heterochromatic repression uncovered previously unappreciated roles for Sir1, a silencing establishment factor, in the maintenance and/or inheritance of silencing. In addition, we identified the sirtuin Hst3 and its histone target as contributors to the stability of the silenced state. These approaches revealed dynamics of a heterochromatin function that have been heretofore inaccessible.

**\*For correspondence:** jrine@berkeley.edu

**Competing interests:** The authors declare that no competing interests exist.

**Reviewing editor**: Daniel E Gottschling, Fred Hutchison Cancer Research Center, United States

## Introduction

Heterochromatin is a heritable, condensed chromatin structure that silences the expression of most genes within or near it. Phenomena such as clonal inheritance of inactivated X-chromosomes in female mammals, as well as position-effect variegation in *Drosophila* and yeast, demonstrate the remarkable ability of cells to propagate heterochromatic repression through mitosis. As an epigenetic state, heterochromatic gene repression provides a means for genetically identical cells to differentiate into stable, distinct cell types. However, despite its significance, little is known about the dynamics of heterochromatic repression and which factors contribute to or disrupt its stability.

In *Saccharomyces cerevisiae*, heterochromatin forms at the silent mating-type loci, *HML* and *HMR*, through the recruitment and subsequent spreading of Sir proteins (*Grunstein and Gasser, 2013*). DNA elements known as the *E* and *I* silencers flank each locus and nucleate complexes of Sir2, Sir3 and Sir4. Sir complexes then deacetylate histones and bind nucleosomes throughout the region, thereby rendering *HML* and *HMR* transcriptionally silenced and largely inaccessible to DNA-interacting proteins. Since each locus contains either **a** or α mating-type information, as does the *MAT* locus, heterochromatic repression of *HML* and *HMR* ensures that the *MAT* genotype is the only determinant of whether haploids mate as **a** or α cells.

Following its initial establishment, Sir-mediated heterochromatin can be maintained through the $G_1$, $G_2$ and M phases and inherited through S phase. Sir2, Sir3 and Sir4 are essential for all aspects of silencing (*Rine and Herskowitz, 1987*). Thus, mutants lacking any of these proteins express *HML* and *HMR* to the level of the transcriptionally active *MAT* locus. In contrast, mutants lacking Sir1 exhibit a bistable silencing phenotype (*Pillus and Rine, 1989*; *Xu et al., 2006*). Within a population of *sir1* cells, *HML* and *HMR* exist in one of two phenotypic states: silenced or expressed. Each state is heritable for multiple cell divisions, demonstrating the epigenetic nature of Sir-mediated heterochromatin and

**eLife digest** A single cell from a plant, an animal or another eukaryote can contain several meters of DNA. In order to fit this length inside the nucleus of the cell, the DNA is wrapped around proteins called histones to form a compact structure known as chromatin.

Chromatin exists in two forms: loosely packed chromatin tends to contain the genes that are expressed in cells, whereas highly compacted chromatin (also called heterochromatin) silences the expression of most nearby genes. Adding small chemical markings to histone proteins can alter how much the chromatin is compacted, and newly formed cells can inherit these markings whenever a cell divides. This enables the new cell to essentially 'remember' which genes were active and which were repressed in the original cell. However, no one has measured how stable heterochromatin is over multiple cell divisions; as such it is unclear if genes that should remain silent are occasionally expressed after a cell has divided a number of times.

Budding yeast is a well-established model for studying heterochromatin. The genome of this single-celled organism has distinct regions of highly compacted chromatin, which are established and maintained by a number of different proteins.

Dodson and Rine have developed a new technique to detect when gene silencing is lost in individual yeast cells, even if the loss of silencing only occurs for a brief period. This approach revealed that genes in supposedly 'silent' heterochromatin were still expressed but only very rarely—in about 1 in every 1000 cell divisions.

Dodson and Rine's findings suggest that when silencing is lost, it is promptly re-established with the aid of a protein called Sir1. In addition, the new technique revealed that Sir1 helps to prevent losses of silencing in the first place. As such, the Sir1 protein appears to have previously unappreciated roles in the maintenance and the inheritance of heterochromatin in dividing yeast cells. Dodson and Rine also discovered that a protein called Hst3—which acts to remove chemical markings from histone proteins—also helps stabilize the silenced state. With this technique in hand, it is now possible to test any molecule, environment, or cellular process for its potential effect on the stability of gene silencing.

inspiring the notion that Sir1 functions in the establishment of silencing, but not the maintenance or inheritance thereafter.

Notably, rare switches occur between the two expression states of *HML* and *HMR* in *sir1* mutants, during which silencing is either lost or established. If Sir1 functioned exclusively in establishment, then losses of silencing should also occur in wild-type cells, yet no such event has been detected. Wild-type expression levels of genes at the *HML* and *HMR* loci are 1000-fold lower than the expression levels of the same genes when at the *MAT* locus, and efforts to detect expression of *HML* and *HMR* by any molecular method have shown the expression signal is indistinguishable from background noise. Moreover, 100% of *MATa* cells respond to α-factor, and diploids homozygous at the *MAT* locus are completely unable to sporulate. Thus, by all previous molecular criteria, the silent mating-type loci are transcriptionally inert. However, heterochromatin undergoes regular exchange of at least some of its structural components with newly synthesized molecules of the same proteins (*Cheng and Gartenberg, 2000*; *Festenstein et al., 2003*; *Cheutin et al., 2003*; *Ficz et al., 2005*) and is subject to perturbations, such as its replication in S phase. These fluctuations in heterochromatin structure imply that either the mechanism of silencing compensates for these changes and flawlessly reassembles each cell cycle, or that there are rare, as yet undetected losses of silencing resulting from heterochromatin dynamics.

To address whether RNA polymerase ever succeeds in transcribing silent chromatin at *HML* and *HMR*, we designed an assay capable of detecting short-lived gene expression with single-cell resolution. By capturing the consequences of transcription with a permanent, heritable mark, we detected transient losses of silencing at *HML* and *HMR* in wild-type cells, characterized the nature of these losses, and identified genetic determinants of heterochromatin stability.

## Results

To determine the stability of gene repression in heterochromatin, we placed the gene encoding the Cre recombinase under control of the *α2* promoter at either *HMLα* or *HMRα* (*Figure 1A*). RNA

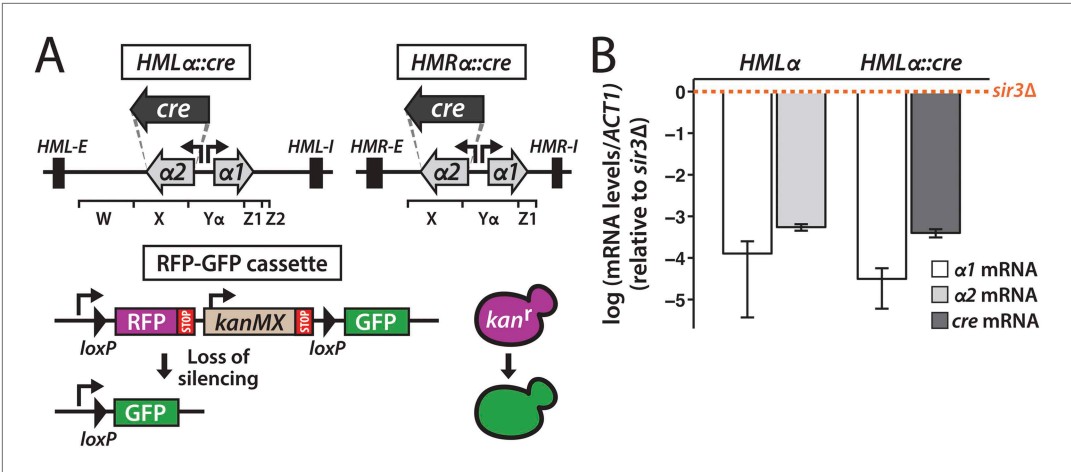

**Figure 1**. Cre-based assay. (**A**) Design of assay to detect short-lived gene expression. The gene coding for the Cre recombinase was integrated at either *HMLα* or *HMRα*, replacing the *α2* coding region, and was expressed from the native *α2* promoter. Black rectangles denote the *E* and *I* silencers of *HML* and *HMR*. The W, X, Yα, Z1 and Z2 regions of the mating-type loci are indicated by brackets. Loss of *cre* silencing induces Cre-mediated recombination of the *loxP* sites (arrowheads) in the RFP-GFP reporter cassette, causing a switch from RFP to GFP expression and a loss of G418 resistance. (**B**) Quantitative RT-PCR analysis of *α1* and *α2* mRNA levels in a strain containing wild-type *HMLα* (JRY9623), and *α1* and *cre* mRNA levels in a strain containing *HMLα::cre* (JRY9625). To determine the fold repression, mRNA levels were also measured in *HMLα* and *HMLα::cre* strains lacking Sir3 (JRY9624 and JRY9626). All strains carried *matΔ* and *hmrΔ* mutations to relieve the repression of *α1* by the a1-α2 heterodimer in *sir3Δ* cells (**Strathern et al., 1981**). mRNA values were normalized to *ACT1* mRNA values. Expression levels are shown relative to the corresponding *sir3Δ* values (set to 1) (orange dotted line). The fold repression of *α2* and *cre* did not significantly differ (p = 0.12; Student's *t* test). Data are means ± standard deviation (SD) (*n* = 3). DOI: 10.7554/eLife.05007.003

measurements made by quantitative RT-PCR showed that *cre* was as repressed as the native *α2* gene at this location (**Figure 1B**). On chromosome V of both the *HMLα::cre* and *HMRα::cre* strains, we integrated a sequence in which two *loxP* sites flanked the *RFP* gene and the selectable drug marker *kanMX* (**Figure 1A**). The *loxP–RFP-kanMX-loxP* sequence resided downstream of the strong *GPD* promoter and upstream of a promoterless *GFP* gene. Thus, cells carrying this RFP-GFP cassette were RFP-positive, drug resistant, and GFP-negative. However, in the event that *cre* repression were lost, the resulting Cre protein could mediate recombination at the *loxP* sites, thereby excising the *RFP* and *kanMX* genes and positioning the *GFP* gene adjacent to the promoter (**Figure 1A**). It should be noted that the recombination event would be essentially irreversible in that the excised DNA lacked an origin of replication and would thereby be lost upon cell division. Therefore, the transient expression of *cre* would trigger a permanent switch to GFP expression and drug sensitivity in that cell and in all of its descendants. By convention, we refer to cells expressing RFP as red, and cells expressing GFP as green.

## Transient losses of silencing detected at *HML* and *HMR*

After plating cells with the *cre* gene at either *HML* or *HMR* on non-selective medium, we imaged the fluorescence of the resulting colonies. We first tested whether the expression of *cre* from the *α2* promoter would produce enough recombinase to cause an efficient RFP-to-GFP switch by adding the Sir2 inhibitor nicotinamide (NAM) to the medium. The nicotinamide-induced derepression of either *HMLα::cre* or *HMRα::cre* resulted in entirely green colonies (**Figure 2A** and **Figure 2—figure supplement 1**). In contrast, cells that contained the RFP-GFP cassette but lacked any source of the *cre* gene grew into colonies that were entirely red (**Figure 2A** and **Figure 2—figure supplement 1**). These observations were consistent with the quantification of GFP fluorescence by flow cytometry (**Figure 2B**). Therefore, expression of the *cre* gene at either *HML* or *HMR* was both necessary and sufficient for the switch from RFP to GFP expression. We then plated *HMLα::cre* and *HMRα::cre* cells on medium lacking nicotinamide to determine the stability of gene silencing under normal conditions. The resulting colonies were predominantly red, indicating the majority of cells maintained the repression of *cre* throughout growth of the colony (**Figure 2A**). Within these colonies, however, we observed discrete green

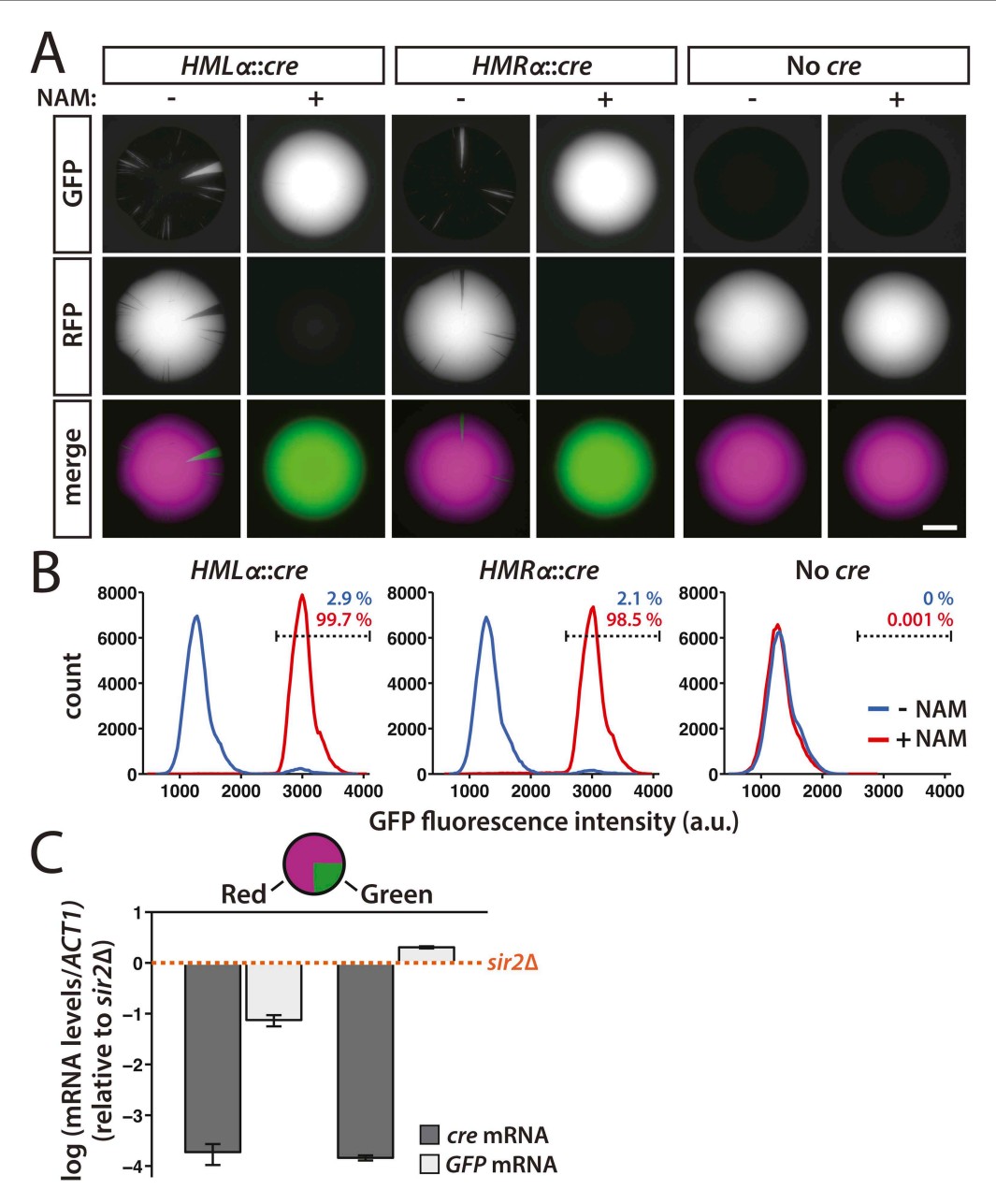

**Figure 2**. Transient transcription of *HML* and *HMR* captured by the Cre-based assay. (**A**) Colonies derived from single cells containing the RFP-GFP cassette in addition to either *HMLα::cre*, *HMRα::cre*, or no source of the *cre* gene (No *cre*). Colonies of each genotype were grown in either the presence or absence of 5 mM nicotinamide (NAM). Additional colonies are shown in *Figure 2—figure supplement 1*. Scale bar, 2 mm. (**B**) Flow cytometry measurements of GFP fluorescence intensity in individual cells from within a single colony. Cells containing the RFP-GFP cassette and either *HMLα::cre*, *HMRα::cre* or no *cre* gene were plated on Complete Supplement Mixture (CSM) −Trp in the presence (red) or absence (blue) of 5 mM NAM. On day 6 of growth, representative colonies of each genotype and condition were resuspended in synthetic complete (SC) medium and grown to early log phase for analysis by flow cytometry. Each distribution represents approximately $10^5$ cells. The percentage of GFP-positive cells shown for each sample was determined by a gate (dotted line) that was set based on the intensity profile of the *HMLα::cre* (+NAM) sample. (**C**) Quantitative RT-PCR measurements of *HMLα::cre* transcription in red and green sectors. To determine whether silencing loss was transient, *cre* and *GFP* mRNA levels were quantified in the GFP-expressing (green) and RFP-expressing (red) regions of a colony of cells containing *HMLα::cre* and the RFP-GFP cassette (JRY9628). Three independent experiments were performed, each on a different colony with a

*Figure 2. Continued on next page*

*Figure 2. Continued*

large green sector. Cells from the red and green regions of each colony were grown to log phase in YPD and harvested for RNA isolation. As a positive control for *HMLα::cre* expression, mRNA levels were also measured in a *sir2Δ* mutant (JRY9633). All mRNA values were normalized to *ACT1* mRNA values. Expression levels are shown relative to the corresponding *sir2Δ* values (set to 1) (orange dotted line). The fold repression of *HMLα::cre* did not significantly differ between the red and green samples (p = 0.43; Student's *t* test). Data are means ± SD (*n* = 3).
The following figure supplement is available for figure 2:

**Figure supplement 1**. GFP fluorescence of additional colonies with either *HMLα::cre*, *HMRα::cre*, or No *cre* grown in the presence or absence of 5 mM nicotinamide (NAM).

sectors, each of which represented a loss-of-silencing event that occurred in a cell at the vertex of the sector (*Figure 2A* and *Figure 2—figure supplement 1*). Although the switch to GFP expression was irreversible, some of the smaller sectors did not extend to the edge of the colony, likely due to a combination of genetic drift (*Hallatschek et al., 2007*) and the nature of three-dimensional growth of the colony. GFP expression was also detectable by flow cytometry in a fraction of *HMLα::cre* and *HMRα::cre* cells (*Figure 2B*). Thus, silent chromatin at *HML* and *HMR* was transcriptionally dynamic, exhibiting losses of silencing below the level of detection by all previous assays.

At least two lines of evidence indicated that these losses of silencing were transient and did not arise from mutations in *SIR* genes. First, *cre* and *GFP* mRNA levels were measured in both red and green regions from within the same colony. If the loss-of-silencing event that resulted in the green sector were temporary, then the level of *HMLα::cre* repression would be indistinguishable between red and green sectors. Alternatively, if the loss of silencing were permanent, then *cre* mRNA would be detectable in cells from the green sector. Consistent with the efficiency of heterochromatin formation, which occurs de novo in approximately 1–2 cell divisions (*Osborne et al., 2009*), *HMLα::cre* repression was fully restored in the descendants of a cell that experienced a loss of silencing (*Figure 2C*). The slight difference in *GFP* expression between wild-type cells from the green sector and *sir2Δ* cells may have been due to the constitutive presence of Cre protein in the *sir2Δ* mutant, which could have potentially disrupted transcription by binding the *loxP* site between the promoter and the *GFP* gene. In addition, the *GFP* mRNA values in the red population likely reflected a low level of GFP expression since colonies were grown in the absence of G418 selection and therefore RFP-to-GFP switches occurred at a low rate. As secondary confirmation that losses of silencing were transient, we have never detected green sectors in colonies of a wild-type strain containing the *GFP* gene under control of the *α2* promoter at *HMLα* (unpublished observations).

The pattern of colony fluorescence revealed the history of individual lapses in silencing, with the size of each green sector corresponding to how early the silencing loss occurred during colony growth. Colonies that were half red/half green reflected a loss of *cre* repression in either the mother or daughter cell of the first division that gave rise to the colony (*Figure 3A*). Therefore, the frequency of half-sectored colonies equaled the rate of RFP-to-GFP switches per cell division. In the *HMLα::cre* strain, the frequency of half-sectored colonies was $1.6 \times 10^{-3}$ (*Table 1* and *Figure 3B*). Thus, for every thousand cells, one to two cells temporarily failed to repress *HMLα*. Silencing of *HMRα* was slightly more stable than *HMLα*, with a lower rate of $7 \times 10^{-4}$ losses per division (p = 0.003; Student's *t* test) (*Table 1* and *Figure 3B*).

## Transient transcription of silent chromatin was restricted to low levels and was captured effectively by the Cre-based assay

In principle, the rates of RFP-to-GFP switches could be compatible with at least three possible distributions of *cre* expression. In one model, *cre* is transcribed at a low level in many cells, but the resulting level of Cre protein is sufficient to catalyze recombination of the RFP-GFP cassette in only a subset of those cells. Alternatively, *cre* transcription could be completely absent in most cells, but in a low fraction of cells, occur at the same level as in a *sir-* mutant. Finally, *cre* transcription may be absent in most cells, and limited even in the small fraction of cells that switch from RFP to GFP expression.

Thus far, efforts to detect transcription at *HML* and *HMR* have relied on quantitative RT-PCR and other population-based assays that report the average level of RNA for all the cells in a sample. However, advances in RNA imaging now enable the quantification of individual transcripts in single

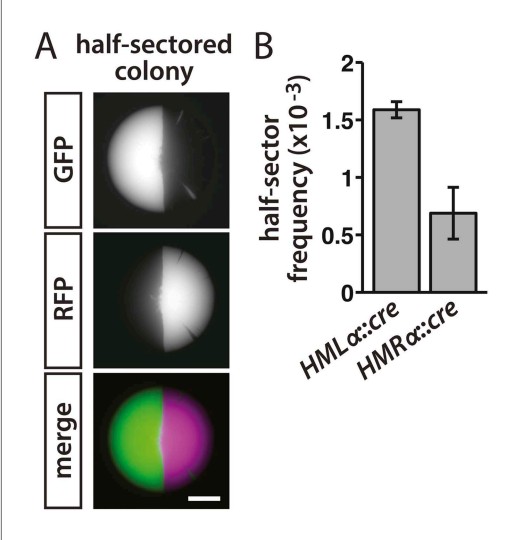

**Figure 3**. Rates of silencing loss as measured by half-sector frequency. (**A**) Example of a half-sectored colony derived from a single cell containing *HMLα::cre* and the RFP-GFP cassette. Scale bar, 1 mm. (**B**) Frequency of half-sectored colonies. The half-sector frequency of a strain containing the *cre* gene at *HMLα* (JRY9628) was significantly different (p = 0.003; Student's *t* test) from the half-sector frequency of a strain containing the *cre* gene at *HMRα* (JRY9629). Data are means ± SD (*n* = 3).

cells (*Itzkovitz and van Oudenaarden, 2011*). Using a version of fluorescence in situ hybridization (FISH) with single-molecule sensitivity (*Raj et al., 2008*), we hybridized fixed cells with two sets of probes distinguished by their fluorophore label (*Figure 4A*). One probe set was specific to the *cre* RNA expressed from *HMLα::cre*. As an internal control to account for sample variation, the other probe set was specific to *KAP104* RNA, which encodes a protein involved in nuclear transport. We measured *cre* and *KAP104* transcripts by imaging and quantifying fluorescent spots, each representative of a single RNA molecule.

As a positive control, we examined *cre* transcription in a mutant lacking Sir4, a protein integral to heterochromatin structure. As expected, *HML* was transcriptionally active in the *sir4Δ* mutant (*Figure 4B,C* and *Figure 4—figure supplement 1*), with a mean of 12 *cre* transcripts per cell (±2 standard error of the mean [SEM] calculated from three independent experiments), and 98% (±1 SEM) of cells containing at least one RNA molecule (*Table 2*). Transcripts of the constitutively active *KAP104* gene were present in 99.0% (±0.3 SEM) of cells (*Table 2*, *Figure 4B,C* and *Figure 4—figure supplement 1*). In concordance with previous studies (*Zenklusen et al., 2008*; *Gandhi et al., 2011*), *KAP104* expression approximated a Poisson distribution, suggesting that most of the cell-to-cell variation in *KAP104*

RNA levels could be explained by stochastic, constitutive transcription. In contrast, the variance of *cre* RNA abundance was large in relation to the mean (*Table 2*), perhaps reflecting an alternative mode of transcription.

In a wild-type strain with *HMLα::cre*, *cre* transcripts were absent in nearly every cell, whereas *KAP104* transcripts were present at a level comparable to that in *sir4Δ* cells (*Figure 4B,C* and *Figure 4—figure supplement 1*). The distribution of *KAP104* expression in wild-type cells was slightly lower than the distribution in the *sir4Δ* mutant, which may be explained by our observation that *sir4Δ* cells were slightly larger than wild type (*Figure 4—figure supplement 2*), as noted before for *sir2Δ* mutants (*Moretto et al., 2013*). Therefore, *HML* was completely silent in the vast majority of cells, indicating the Cre-based assay was sensitive to rare events of transcription. Although we detected an apparent *cre* signal in $4.2 \times 10^{-3}$ ($\pm 0.7 \times 10^{-3}$ SEM) wild-type cells, we also detected an apparent *cre* signal at a frequency of $3 \times 10^{-3}$ ($\pm 1 \times 10^{-3}$ SEM) in cells that lacked the *cre* gene, which were imaged to control for hybridization specificity (*Table 2*, *Figure 4B,C* and *Figure 4—figure supplement 1*). The wild-type strain and the negative control strain were distinguishable, however, by the frequency of cells containing more than one spot of *cre* signal (*Table 2*). We observed five wild-type cells across all three replicates (13,695 wild-type cells total) that contained more than one *cre* spot, ranging from 2 to 4 spots per cell, whereas we never detected more than one *cre* spot per cell in the negative control strain (13,722 No-*cre* cells total). Together, the RNA FISH analysis and the Cre-based assay converged on the same striking conclusion: the low rate of RFP-to-GFP switches reflected the complete absence of *HMLα::cre* transcription in the vast majority of cells.

There was no evidence of any wild-type cells containing *cre* RNA levels typical of *sir4Δ* cells, suggesting that in the rare cells that lost silencing, the duration of transcription was short, presumably due to continuous nucleation of silent chromatin by Sir1 and other factors. Silencing in individual *sir1* cells was previously measured by indirect bioassays (*Pillus and Rine, 1989*; *Xu et al., 2006*), and with respect to transcription, the expression levels of *HML* and *HMR* underlying the two phenotypically distinct cell types in a *sir1* population have never been established. RNA FISH allowed us to evaluate

**Table 1.** Frequency of half-sectored colonies for each strain that showed a sectoring phenotype

| Figure | Strain | Relevant genotype | Half-sector frequency |
|---|---|---|---|
| 3 | JRY9628 | *matΔ HMLα::cre* | 0.00158 ± 0.00007 |
| | JRY9629 | *matΔ HMRα::cre* | 0.0007 ± 0.0002 |
| 5 | JRY9739 | *MATa HMLα::cre sir1Δ* | 0.055 ± 0.007 |
| | JRY9740 | *MATa HMRα::cre sir1Δ* | 0.061 ± 0.002 |
| 6 | JRY9731 | *MATα/matΔ HMLα/HMLα::cre* | 0.00037 ± 0.00004 |
| | JRY9732 | *MATα/matΔ HMLα/HMLα::cre sir1Δ/SIR1* | 0.0007 ± 0.0002 |
| | JRY9734 | *MATα/matΔ HMLα/HMLα::cre sir3Δ/SIR3* | 0.0016 ± 0.0005 |
| | JRY9735 | *MATα/matΔ HMLα/HMLα::cre sir4Δ/SIR4* | 0.0013 ± 0.0003 |
| 7 | JRY9636 | *matΔ HMLα::cre hst3Δ* | 0.0111 ± 0.0005 |
| | JRY9639 | *MATa HMLα::cre hht1-hhf1Δ hht2-hhf2Δ [HHT2-HHF2]* | 0.0033 ± 0.0004 |
| | JRY9640 | *MATa HMLα::cre hht1-hhf1Δ hht2-hhf2Δ [hht2K56R-HHF2]* | 0.0111 ± 0.0008 |
| | JRY9641 | *MATa HMLα::cre hht1-hhf1Δ hht2-hhf2Δ [hht2K56Q-HHF2]* | 0.009 ± 0.002 |
| | JRY9736 | *MATa HMLα::cre hst3Δ hht1-hhf1Δ hht2-hhf2Δ [HHT2-HHF2]* | 0.0096 ± 0.0008 |
| | JRY9737 | *MATa HMLα::cre hst3Δ hht1-hhf1Δ hht2-hhf2Δ [hht2K56R-HHF2]* | 0.009 ± 0.001 |
| | JRY9738 | *MATa HMLα::cre hst3Δ hht1-hhf1Δ hht2-hhf2Δ [hht2K56Q-HHF2]* | 0.0105 ± 0.0004 |

All values represent the mean of three independent experiments ± standard deviation. Full dataset is shown in **Table 1—source data 1**. All strains contained a copy of the RFP-GFP cassette. See **Supplementary file 1** for full genotypes.

**Source data 1**. Half-sector frequencies from three independent experiments.

*sir1Δ* cells at the level of transcription on a per cell basis to determine whether these two states were molecularly equivalent to the levels of *HML* expression seen in *sir4Δ* and wild-type cells. In the absence of Sir1, 46% (±7 SEM) of cells expressed *HML* (**Table 2**), exhibiting a range in *cre* RNA levels that resembled the *sir4Δ* mutant (**Figure 4B,C** and **Figure 4—figure supplement 1**). The remaining 54% of *sir1Δ* cells lacked detectable *cre* RNA molecules, but expressed *KAP104* at normal levels (**Figure 4B,C** and **Figure 4—figure supplement 1**). Therefore, in approximately half of all cells lacking Sir1, *HML* was fully repressed, whereas the other half expressed *HML* at levels indistinguishable from those in a *sir4Δ* mutant.

## Effects of Sir protein availability on the dynamics of silencing

Although Sir1 was defined by its role in the establishment of silencing, and is not required for maintenance or inheritance, the possibility that it contributes to these other processes has never been directly tested. Such a test would require a comparison of the rate of silencing loss in wild-type cells to the rate of silencing loss in the population of *sir1* cells in which *HML* was silenced. Whereas past measurements in wild-type cells were confounded by the role of Sir1 in establishment (**Xu et al., 2006**), the Cre-based assay is sensitive enough to record losses of silencing before re-establishment occurs.

To determine whether Sir1 contributes to the maintenance or inheritance of silencing, we sporulated a *sir1* hemizygote containing *HMLα::cre* and the RFP-GFP cassette and then imaged the fluorescence of colonies derived from the meiotic products. If the rate of silencing loss were unaffected by the absence of Sir1, then the sectoring patterns would be indistinguishable between colonies arising from *SIR1* cells and colonies arising from *sir1Δ* cells in which *HMLα::cre* was repressed. However, green sectors were notably more abundant in the colonies of spores lacking Sir1 (**Figure 5A**, **Figure 5—figure supplement 1**). Moreover, the frequency of half-sectored colonies in a *sir1Δ* haploid was 35-fold above the frequency of half-sectored colonies in a wild-type haploid (**Table 1** and **Figure 5B**), indicating that cells lost silencing at *HML* at a strikingly higher rate when Sir1 was absent. In strains containing *HMRα::cre* instead of *HMLα::cre*, *sir1Δ* colonies showed elevated sectoring and a higher abundance of half sectors in comparison to wild-type colonies, as well (**Table 1** and **Figure 5**). Thus, beyond its role in establishment, Sir1 contributed to either the maintenance or inheritance of silencing at both *HMLα*

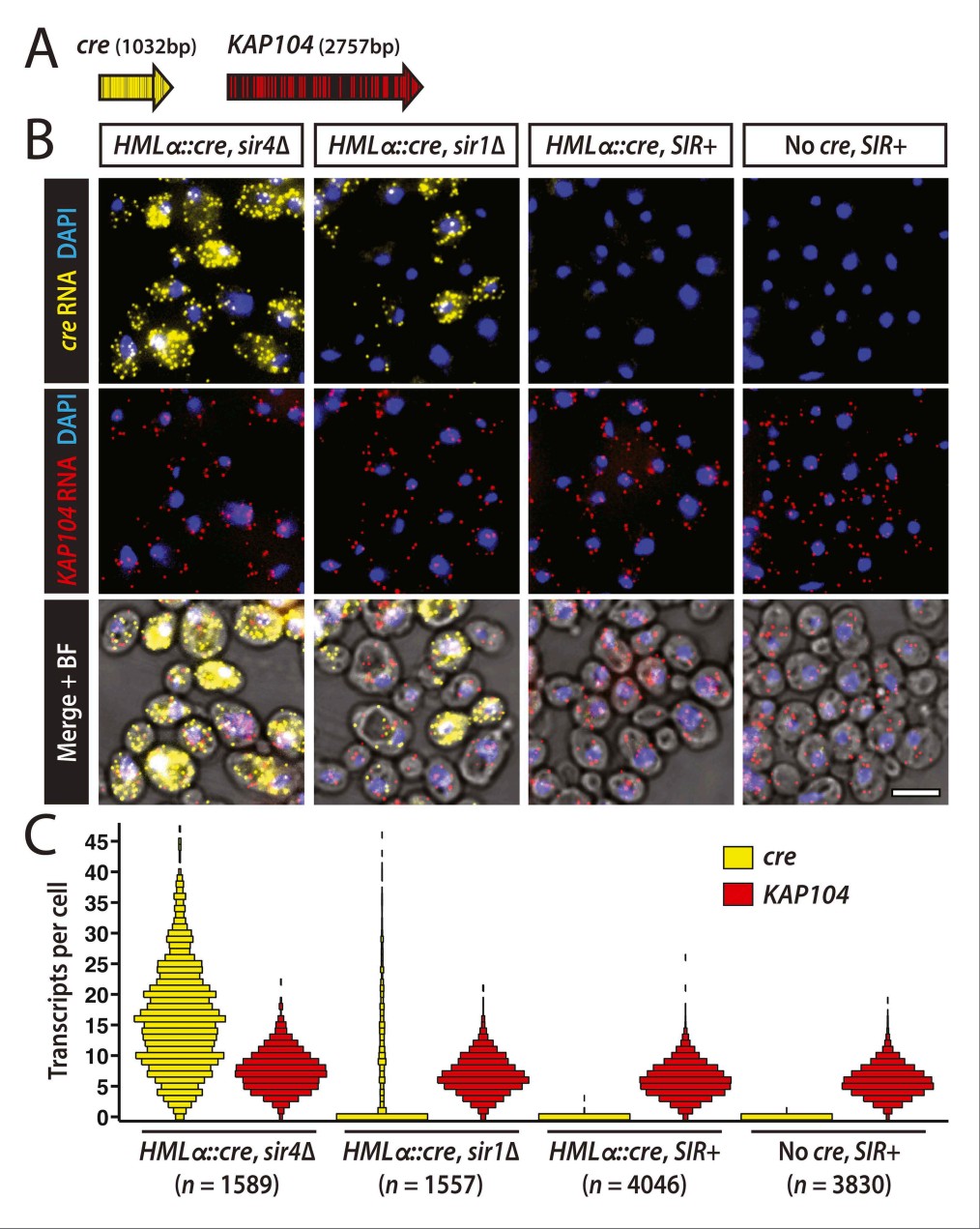

**Figure 4**. Single-molecule imaging of *HMLα::cre* transcription. (**A**) Schematic of the *cre* and *KAP104* coding sequences (black arrows), with colored lines representing sites targeted by the 20-nucleotide FISH probes (see *Supplementary file 4* for probe sequences). To detect both RNA sequences in the same cell, the *cre*-specific and *KAP104*-specific probes were differentially labeled. (**B**) Single-molecule RNA FISH of *cre* (yellow) and *KAP104* (red) transcripts shown as maximum-intensity projections of z-stacks. DNA was stained with 4',6-diamidino-2-phenylindole (DAPI) (blue). Merged images were superimposed on the corresponding brightfield (BF) micrographs. Larger fields are shown in *Figure 4—figure supplement 1*. Scale bar, 5 µm. (**C**) Distributions of *cre* and *KAP104* RNA abundance determined by FISH. Each violin plot was normalized so that maximum bin width was the same across all samples. Data shown are from a representative replicate.

The following figure supplements are available for figure 4:

**Figure supplement 1**. Imaging of *HMLα::cre* and *KAP104* transcription.

**Figure supplement 2**. Boxplots of cell widths measured using brightfield images from the RNA FISH replicate shown in *Figure 4C*.

**Table 2.** Summary of single-molecule RNA FISH statistics

| RNA | Genotype | Mean spots per cell | Variance | % Cells with >0 spots | % Cells with >1 spot |
|-----|----------|---------------------|----------|-----------------------|----------------------|
| *cre* | *HMLα::cre, sir4Δ* | 12 ± 2 | 53 ± 9 | 98 ± 1 | 95 ± 2 |
| | *HMLα::cre, sir1Δ* | 5 ± 1 | 58 ± 13 | 46 ± 7 | 41 ± 7 |
| | *HMLα::cre, SIR+* | <0.005 | – | 0.42 ± 0.07 | 0.04 ± 0.01 |
| | No *cre, SIR+* | <0.003 | – | 0.3 ± 0.1 | 0 |
| KAP104 | *HMLα::cre, sir4Δ* | 7.3 ± 0.4 | 11 ± 1 | 99.0 ± 0.3 | 97.2 ± 0.8 |
| | *HMLα::cre, sir1Δ* | 7.1 ± 0.2 | 10.8 ± 0.9 | 99.28 ± 0.04 | 97.5 ± 0.1 |
| | *HMLα::cre, SIR+* | 6.1 ± 0.2 | 7.8 ± 0.3 | 99.1 ± 0.2 | 96.5 ± 0.8 |
| | No *cre, SIR+* | 5.9 ± 0.5 | 7.1 ± 0.5 | 99.0 ± 0.7 | 96 ± 2 |

All values represent the mean of three independent experiments ± standard error of the mean. Full dataset is shown in **Table 2—source data 1**.

**Source data 1**. Single-molecule RNA FISH values from three independent experiments.

and *HMRα*. Consistent with the RNA FISH measurements, which revealed a large population of *sir1Δ* cells that lacked detectable *cre* transcripts (**Table 2** and **Figure 4C**), many of the cells within the *sir1Δ* colonies remained red (**Figure 5A**). Therefore, *sir1Δ* cells that were in the silenced state exhibited the same level of repression as wild-type cells, but transitioned to a transcriptionally active state more often than wild-type cells.

Sir1 facilitates the recruitment of Sir2, Sir3 and Sir4 and therefore helps maintain a local concentration of heterochromatin components at *HML* and *HMR*. To test whether the dynamics of silencing were sensitive to the availability of Sir proteins, *HMLα::cre* expression was monitored in diploids hemizygous for individual *SIR* genes. While measuring silencing loss in diploids, we noticed that a wild-type diploid with one copy of *HMLα::cre* and the RFP-GFP cassette showed considerably reduced colony sectoring and a fourfold decrease (p = 1 × 10$^{-5}$; Student's *t* test) in the frequency of half-sectored colonies in comparison to an isogenic haploid (**Table 1**), indicating a ploidy effect on the rate of RFP-to-GFP switching. Compared to the wild-type diploid, colonies of the *sir1*, *sir3* and *sir4* hemizygotes showed a modest yet consistent increase in the number of green sectors (**Figure 6A** and **Figure 6—figure supplement 1**) and a higher occurrence of half sectors (**Table 1** and **Figure 6B**). Therefore, reducing the dose of the genes encoding Sir3 or Sir4, two structural components of heterochromatin, or Sir1, a protein that recruits these components, rendered silencing at *HML* less stable. Altered gene dosage of *SIR1* and *SIR4* also affects the silencing of a sensitized *HMR::ADE2* allele (**Sussel et al., 1993**). Hemizygosity for *SIR2* had no perceivable impact on colony sectoring (**Figure 6A** and **Figure 6—figure supplement 1**).

## A novel role for the sirtuin Hst3 in the stabilization of silencing at *HML*

Sir2 is the only member of the sirtuin family of NAD$^+$-dependent deacetylases previously shown to have a role in silencing at *HML* or *HMR*. As measured by mating efficiency, silencing is indistinguishable between wild type and mutants lacking any of the four *HST* genes (**Brachmann et al., 1995**; **Yang et al., 2008**). However, Hst3 and Hst4 have been shown to contribute to the silencing of subtelomeric genes (**Brachmann et al., 1995**; **Yang et al., 2008**), as well as silencing of a *URA3* reporter in a plasmid-borne *HMR* cassette (**Grünweller and Ehrenhofer-Murray, 2002**). Since nicotinamide inhibits all five sirtuins in yeast (**Imai et al., 2000**; **Landry et al., 2000a, 2000b**; **Smith et al., 2000**; **Tanner et al., 2000**; **Tanny and Moazed, 2001**), we tested whether other sirtuins, in addition to Sir2, were involved in the nicotinamide-induced loss of silencing observed in **Figure 2**. As expected, colonies of cells with *HMLα::cre* and the RFP-GFP cassette turned completely green in the absence of Sir2, a protein essential for the nucleation and maintenance of heterochromatin (**Figure 7A** and **Figure 7—figure supplement 1A**). The colonies of *hst1Δ*, *hst2Δ* and *hst4Δ* mutants exhibited sectoring patterns that were indistinguishable from wild-type colonies (**Figure 7A** and **Figure 7—figure supplement 1A**). In contrast, the *hst3Δ* mutant showed a striking increase in colony sectoring and a sevenfold greater

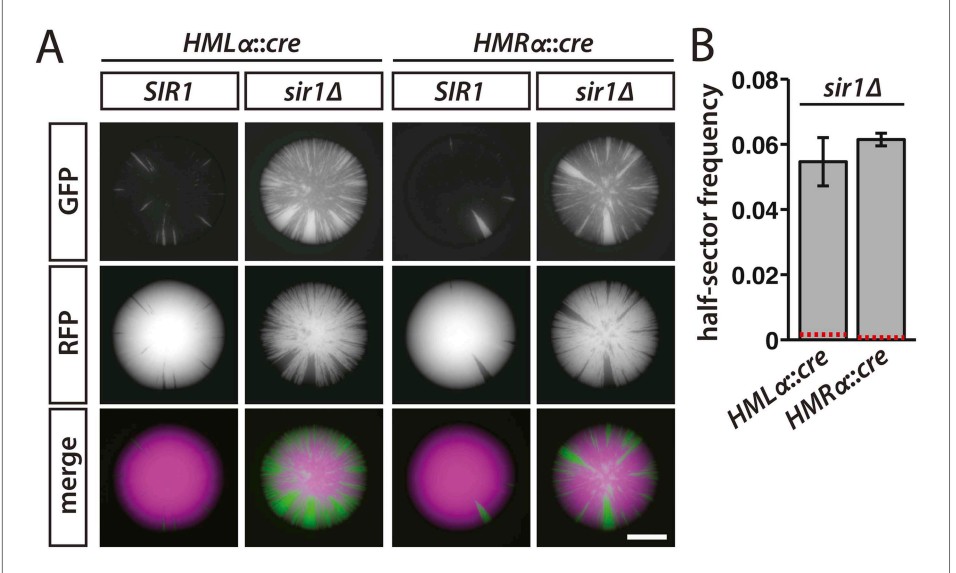

**Figure 5**. Increased silencing loss in cells lacking Sir1. (**A**) Fluorescence of colonies derived from *SIR1* and *sir1Δ* cells containing the RFP-GFP cassette and either *HMLα::cre* or *HMRα::cre*. All colonies shown were grown from *MATa* spores of JRY9729 and JRY9730 tetrad dissections. Additional colonies are shown in *Figure 5—figure supplement 1*. Scale bar, 1 mm. (**B**) Frequencies of half-sectored colonies for haploid strains carrying the *sir1Δ* mutation and either *HMLα::cre* (JRY9739) or *HMRα::cre* (JRY9740). For both *HMLα::cre*- and *HMRα::cre*-containing strains, the half-sector frequency of the *sir1Δ* mutant was significantly higher (p = $2 \times 10^{-4}$ for *HMLα::cre*, p = $7 \times 10^{-7}$ for *HMRα::cre*; Student's *t* test) than the half-sector frequency of wild type (red dotted lines represent the rates shown in *Figure 3B*). Data are means ± SD (*n* = 3).
The following figure supplement is available for figure 5:

**Figure supplement 1**. GFP fluorescence of additional *SIR1* and *sir1Δ* colonies carrying the RFP-GFP cassette and either *HMLα::cre* or *HMRα::cre*.

frequency of half-sectored colonies compared to wild type (*Table 1*, *Figure 7A,B* and *Figure 7—figure supplement 1A*). Thus, the Hst3 deacetylase contributed to the stability of silenced chromatin at *HML*. Whereas silencing defects have not been detected at subtelomeric genes in either the *hst3Δ* or *hst4Δ* single mutant, the *hst3Δ hst4Δ* double mutant shows a measurable phenotype (*Brachmann et al., 1995*; *Yang et al., 2008*). Therefore, it is likely that the deletion of *HST4* would enhance the sectoring phenotype of the *hst3Δ* mutant shown in *Figure 7A*.

Hst3 regulates the deacetylation of acetylated lysine 56 on histone H3 (H3 K56-ac) (*Celic et al., 2006*; *Maas et al., 2006*; *Yang et al., 2008*). To determine whether Hst3 stabilized silencing through deacetylation of H3 K56-ac or through deacetylation of yet undiscovered substrates, we measured the effects of amino acid substitutions that either mimic (K56Q) or prevent (K56R) the acetylation of H3 K56. Although these histone gene mutations were previously analyzed for their effects on silencing, the outcome varies depending on the locus and the assay (*Hyland et al., 2005*; *Xu et al., 2007*; *Miller et al., 2008*; *Yu et al., 2011*). In our strains, with *cre* under the native *α2* promoter at *HML*, colonies of the K56Q mutant showed a dramatic increase in sectoring, whereas the K56R mutant was similar to wild type (*Figure 7C* and *Figure 7—figure supplement 1B*). Consistent with the possibility that the *hst3Δ* silencing defects were due to an increase in acetylated histone H3 K56, the sectoring phenotypes suggested that neutralizing the positive charge at this position on histone H3 impaired the stability of repression. By half-sector analysis, however, both the K56Q and K56R mutants showed a significant increase (p = $3 \times 10^{-3}$ and p = $9 \times 10^{-5}$, respectively; Student's *t* test) in the rate of silencing loss compared to wild type (*Table 1* and *Figure 7D*). The relatively high frequency of half-sectored colonies in the histone H3 K56R mutant was unexpected due to the similarity in sectoring patterns between histone H3 K56R mutant colonies and wild-type colonies (*Figure 7C* and *Figure 7—figure supplement 1B*). However, the histone H3 K56R substitution has previously been shown to decrease

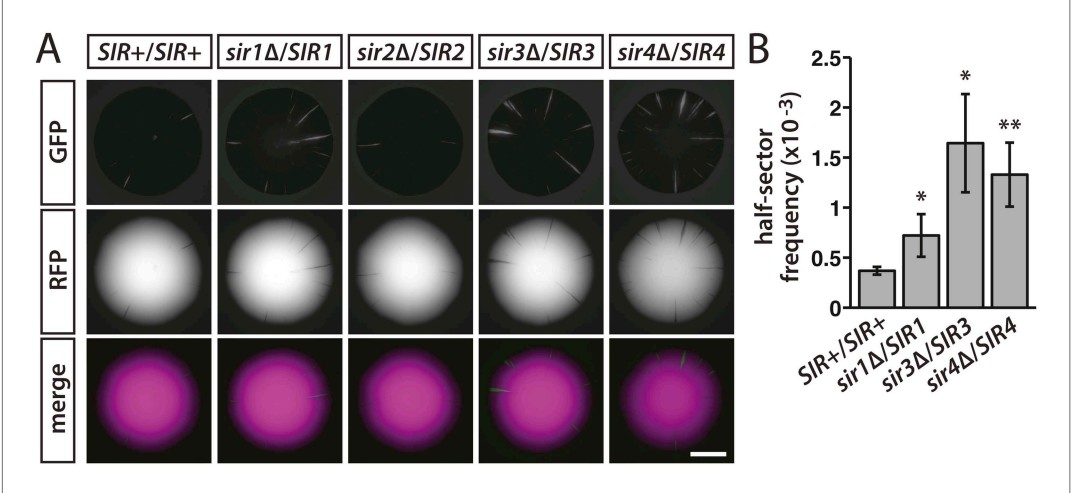

**Figure 6**. Stability of silencing at *HML* in diploids hemizygous for individual *SIR* genes. (**A**) Fluorescence of colonies derived from single diploid cells containing one copy of *HMLα::cre*, one copy of the RFP-GFP cassette, and the indicated *SIR* genotypes. Additional colonies are shown in *Figure 6—figure supplement 1*. Scale bar, 2 mm. (**B**) Frequencies of half-sectored colonies for a wild-type diploid and different *sir* hemizygotes. The half-sector frequencies of the *sir1*, *sir3* and *sir4* hemizygotes were significantly higher (*p < 0.05, **p < 0.01; Student's *t* test) than the half-sector frequency of the wild-type diploid. Data are means ± SD (*n* = 3). The frequency of half-sectored colonies was not determined for the *sir2* hemizygote.

The following figure supplement is available for figure 6:

**Figure supplement 1**. GFP fluorescence of additional colonies derived from single diploid cells hemizygous for individual *SIR* genes.

---

silencing under various conditions (*Xu et al., 2007*; *Yu et al., 2011*). Differences between sectoring patterns and half-sector frequencies are considered further in the 'Discussion'.

Cells expressing the H3 K56Q mutant produced similar patterns of colony sectoring regardless of whether Hst3 was present (*Figure 7C* and *Figure 7—figure supplement 1B*). This phenotype was slightly less severe than that of *hst3Δ* cells expressing wild-type H3 K56 (*Figure 7C* and *Figure 7— figure supplement 1B*), perhaps because glutamine was not quite as disruptive to silencing as acetylated lysine at this residue. In addition, the H3 K56R substitution in *hst3Δ* cells restored silencing to wild-type levels, as determined by overall colony sectoring (*Figure 7C* and *Figure 7—figure supplement 1B*). Therefore, blocking the acetylation of residue 56 suppressed the *hst3Δ* sectoring phenotype. Collectively, the sectoring patterns suggested that Hst3 promoted the stability of silencing through the deacetylation of histone H3 K56-ac. Whereas the deletion of *HST3* caused a threefold increase in the half-sector frequency of wild type (p = 3 × 10$^{-4}$; Student's *t* test), it did not significantly affect the half-sector frequency of the H3 K56Q mutant (p = 0.3; Student's *t* test) or the surprisingly high half-sector frequency of the H3 K56R mutant (p = 0.06; Student's *t* test) (*Table 1* and *Figure 7D*). Thus, the state of histone H3 K56 affected the stability of silencing at *HML*, and amino acid substitutions of this residue were epistatic to the *hst3Δ* mutation.

Colonies derived from cells in which the only source of histone H3–H4 genes was a plasmid-borne copy of *HHT2-HHF2* exhibited more sectoring than colonies of cells containing both *HHT1-HHF1* and *HHT2-HHF2* at their native chromosomal loci (*Figure 7—figure supplement 2*). Assuming the decrease in gene copy number led to a decrease in protein abundance, silencing was likely affected by levels of histones H3 and H4 that were either limiting or improperly balanced with the levels of histones H2A and H2B. Consistent with these possibilities, histone dosage has previously been shown to affect various functions of heterochromatin (*Moore et al., 1979*; *Moore et al., 1983*; *Venditti et al., 1999*).

In principle, the effect of histone H3–H4 gene dosage should sensitize silencing in all genetic backgrounds. Whereas the reduction from two copies of the histone H3–H4 gene pair to one copy did enhance the sectoring phenotype of *hst3Δ* colonies (compare panels A and C in *Figure 7* and

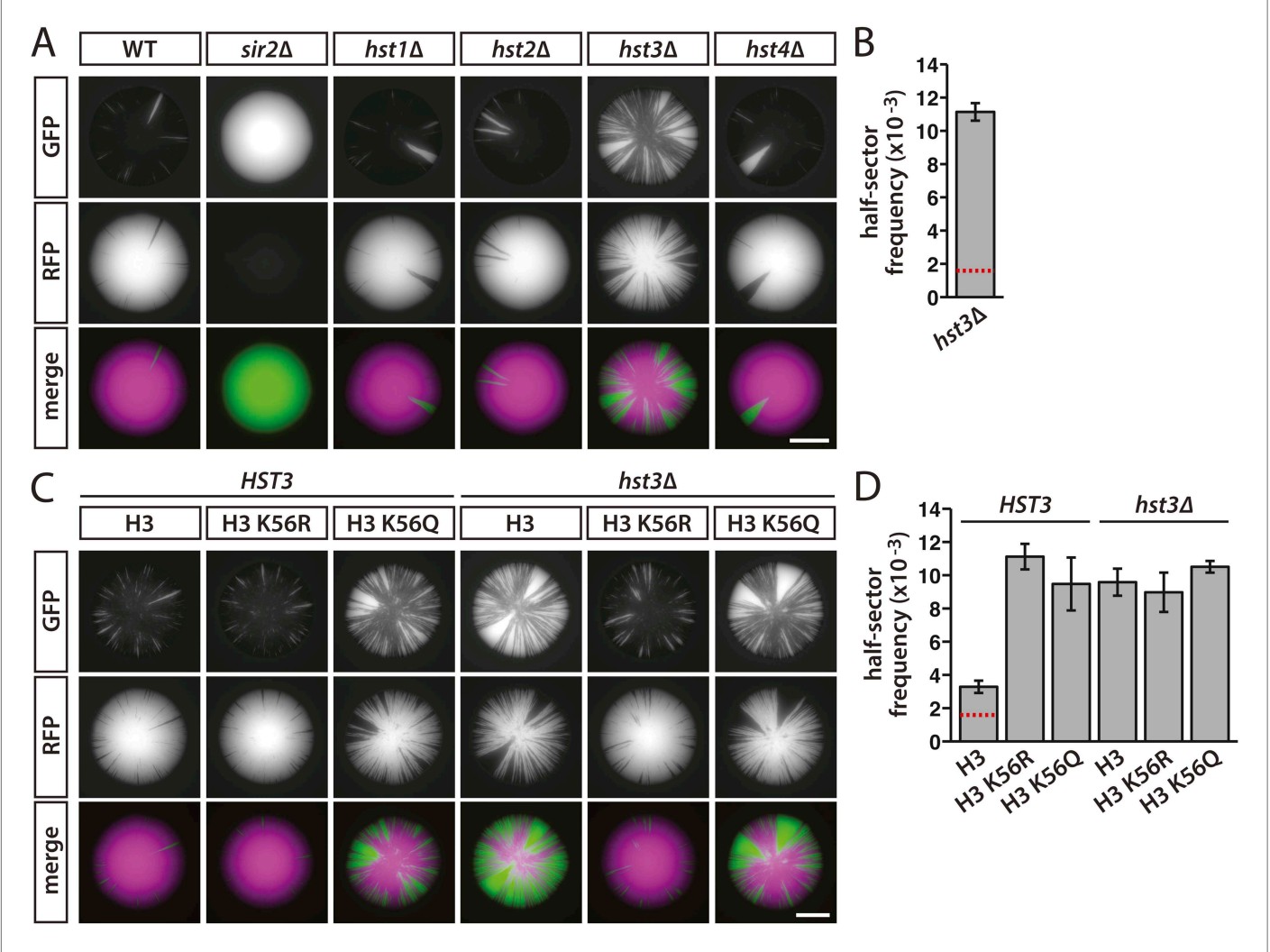

**Figure 7**. Effects of sirtuin gene deletions and histone H3 K56 or K56Q substitutions on the stability of silencing at *HML*. (**A**) Fluorescence of colonies derived from single cells containing *HMLα::cre*, the RFP-GFP cassette, and the indicated deletions. Additional colonies are shown in *Figure 7—figure supplement 1A*. Scale bar, 2 mm. (**B**) Frequency of half-sectored colonies for the *hst3Δ* mutant shown in panel **A**. The half-sector frequency of the *hst3Δ* mutant was significantly greater (p = 6 × 10⁻⁶; Student's *t* test) than the half-sector frequency of wild type (red dotted line represents the rate shown in *Figure 3B*). Data are means ± SD (*n* = 3). The frequency of half-sectored colonies was not determined for the *hst1Δ*, *hst2Δ* or *hst4Δ* mutant. (**C**) Fluorescence of colonies derived from single cells containing *HMLα::cre*, the RFP-GFP cassette, *hht1-hhf1Δ* and *hht2-hhf2Δ* mutations, and a plasmid-borne copy of either *HHT2-HHF2*, *hht2K56R-HHF2* or *hht2K56Q-HHF2*. Additional colonies are shown in *Figure 7—figure supplement 1B*. Scale bar, 2 mm. (**D**) Frequency of half-sectored colonies for the genotypes shown in panel **C**. The half-sector frequency of the wild-type strain containing only one copy of the histone H3–H4 gene pair was significantly higher (p = 0.001; Student's *t* test) than the half-sector frequency of the strain shown in *Figure 3B* containing both copies of the histone H3–H4 gene pair (red dotted line) (see *Figure 7—figure supplement 2*). Data are means ± SD (*n* = 3).

The following figure supplements are available for figure 7:

**Figure supplement 1**. Additional colonies showing the effects of sirtuin gene deletions and histone H3 K56 substitutions on the stability of silencing at *HML*.

**Figure supplement 2**. Histone H3–H4 gene dosage effect on silencing at *HML*.

panels A and B in *Figure 7—figure supplement 1*), it did not increase the half-sector frequency in the *hst3Δ* background as it did in the wild-type background (*Table 1* and *Figure 7B,D*). Therefore, the degree to which histone H3–H4 gene dosage affected silencing, at least during the first cell division of colony growth, depended on the genetic background.

## Live-cell imaging revealed a variety of switching patterns

Whereas colony sectors revealed the history of heterochromatin dynamics, we sought to also capture these events in real time at single-cell resolution to determine whether observable losses of silencing exhibited a fixed pattern or were more stochastic. Therefore, we monitored the fluorescence of cells containing *HMLα::cre* and the RFP-GFP cassette as they divided in a chamber supplied with fresh medium. Most cells remained red throughout the duration of the time course. Consistent with the sectors observed within colonies, rare RFP-to-GFP switches occurred in a subset of the lineages (see *Video 1* for an example). The rate of switching observed with live-cell imaging was within 10% of the rate of switching determined by the half-sector assay.

The time lag between the expression of *cre* and the ultimate maturation of GFP limits the temporal resolution from such analyses. Nonetheless, a survey of the pattern of switch events allowed some useful inferences. For each RFP-to-GFP switch, we determined whether cells were unbudded, small-budded, or large-budded at the moment when GFP expression was initially detected. We observed a fluorescent switch in 24 different cells, nine of which were unbudded and thus in the G1 phase during the onset of GFP expression (*Figure 8A,D*). Multiple occurrences of both patterns 1 and 2 (*Figure 8D*) indicated that silencing loss followed by recombination was not restricted to either the mother or daughter cell (Given that pattern 3 was supported by only one example, we refrained from offering an interpretation of this event).

Of the remaining cells that switched, three cells had small buds when GFP fluorescence was first detected and 12 had large buds (*Figure 8D*). All three small-budded cells gave rise to mothers and daughters that both remained green (pattern 6) (*Figure 8B,D*). Since the bud emerges during early S phase, and there were no cases of only the mother or daughter cell remaining green following the onset of GFP expression in a small-budded cell (patterns 4 and 5), it was likely that recombination of the *loxP* sites occurred prior to DNA replication. Alternatively, recombination may have occurred shortly after DNA replication at both newly synthesized copies of the RFP-GFP cassette. In contrast, 8 of the 12 cells that expressed detectable GFP as large-budded cells gave rise to mother-daughter pairs in which only the mother or daughter remained green (patterns 7 and 8) (*Figure 8C,D*), indicating that Cre catalyzed recombination after S phase and acted on only one of the two RFP-GFP cassettes. Switches that followed patterns 7 and 8 also demonstrated that the time between recombination, which occurred some point after DNA replication, and GFP detection, which occurred when a large bud was present, spanned only a fraction of a complete cell cycle.

For three of the RFP-to-GFP switches, we observed a second switch in the daughter cell that arose from the immediately preceding cell division (*Figure 8B,D* and *Figure 8—figure supplement 1*). These paired switches, which occurred either simultaneously or within 40 min of each other, could have reflected either two loss-of-silencing events in directly related cells, or rather one loss-of-silencing event that happened when the mother cell and its daughter still shared a cytoplasm, assuming there was perdurance of the Cre protein in both cells and a delay in recombination. These patterns are considered further in *Figure 8—figure supplement 1* and the 'Discussion'.

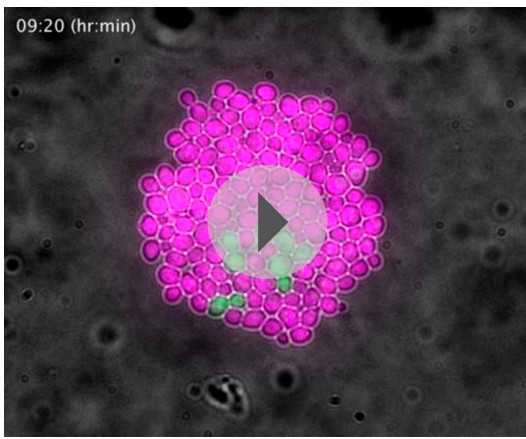

**Video 1**. RFP-to-GFP switch captured in real time. Time-lapse imaging of wild-type cells containing *HMLα::cre* and the RFP-GFP cassette. Images were taken every 10 min for 13 hr.

## Discussion

Several lines of evidence establish the dynamic nature of heterochromatin structure. The incorporation of newly synthesized Sir3 into silenced chromatin during $G_1$-phase arrest suggests that at least some Sir3 binding to nucleosomes is transient (*Cheng and Gartenberg, 2000*). Similarly, fluorescence-recovery-after-photobleaching experiments have shown that heterochromatin protein 1 (HP1), a structural component of heterochromatin in other eukaryotes, can rapidly associate and dissociate from heterochromatic loci (*Cheutin et al., 2003*; *Festenstein et al., 2003*).

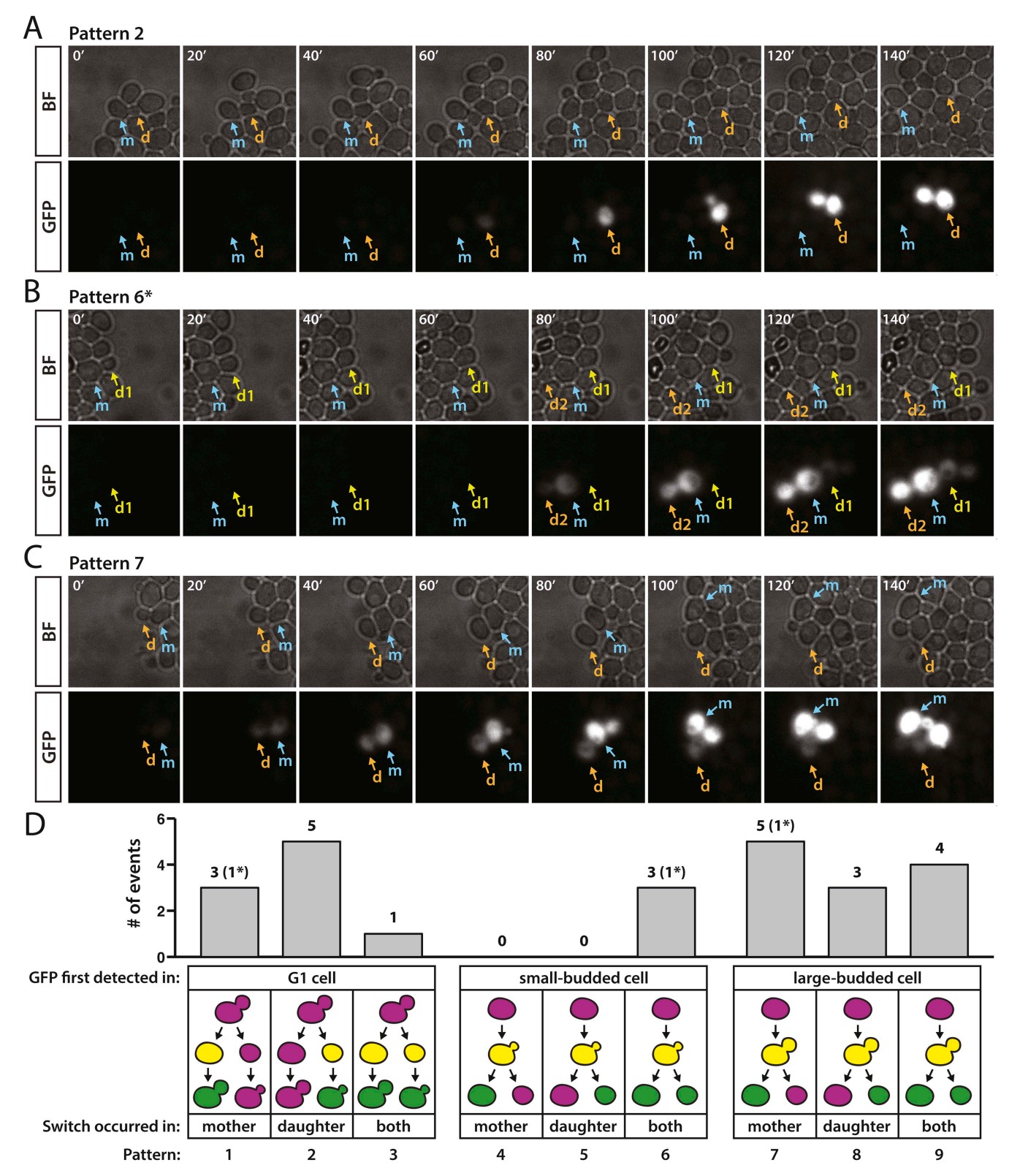

**Figure 8**. Patterns of RFP-to-GFP switches observed in real time. (**A–C**) Brightfield (BF) and GFP fluorescence montages (shown in 20-min intervals, 140 min total) showing different switching patterns of wild-type cells containing *HMLα::cre* and the RFP-GFP cassette. Labeled arrows track the mother (m) and daughter (d) cells of interest. (**A**) GFP expression first detected in an unbudded daughter cell. (**B**) GFP expression first detected in a

*Figure 8. Continued on next page*

*Figure 8. Continued*

small-budded cell. Both the mother (m) and daughter (d2) cells remained GFP-positive (pattern 6). *A second RFP-to-GFP switch occurred in the daughter (d1) of the preceding cell division (see *Figure 8—figure supplement 1*, pedigree B). (**C**) GFP expression first detected in a large-budded cell. Only the mother (m) cell remained GFP-positive. (**D**) Distribution of different switch patterns. RFP-to-GFP switches were initially categorized by bud morphology at the time that GFP expression was first detected (depicted as yellow cells in the cartoons since RFP was still present). Events in each bud category were then classified according to whether the mother cell, daughter cell, or both underwent recombination of the RFP–GFP cassette, as determined by whether they continued to express GFP (depicted as green cells in the last phase of the pedigree cartoons). Asterisks denote the number of events within a particular category that were associated with a second switch in a directly related cell (see *Figure 8—figure supplement 1*).

The following figure supplement is available for figure 8:

**Figure supplement 1**. Cartoons of lineages that showed two RFP-to-GFP switches in directly related cells.

However, the consequences of these structural fluctuations on the underlying functions of heterochromatin remain poorly understood. Here, we showed that heterochromatin dynamics had a functional impact on the transcriptional repression imparted by Sir proteins. Moreover, we have established a route to a comprehensive understanding of all factors influencing heterochromatin dynamics.

## Inherent dynamics of heterochromatic gene silencing

At a low rate, the heterochromatic loci *HML* and *HMR* underwent losses of silencing that had escaped detection by all previous efforts, due to the combination of their rarity and transience. Independent observations indicated that silencing was promptly re-established following its loss. First, *HMLα::cre* was fully silenced in the descendants of cells that switched to GFP expression, showing that *cre* expression was only temporary. Secondly, single-molecule RNA FISH did not detect any wild-type cells expressing *HMLα::cre* to the average level of *sir4Δ* cells, implying that when silencing was lost in wild-type cells, it was restored before *cre* transcription could reach that level. These data, taken together with the inability of less sensitive assays to capture expression of *HML* or *HMR*, suggested that losses of silencing were rare and short-lived.

The detection of silencing loss in a subset of wild-type cells supported a model where Sir proteins and the transcriptional machinery constantly compete for access to the silent mating-type loci. Such a concept is consistent with the dynamic exchange of structural components in heterochromatin, the continuous recruitment of Sir proteins by the silencers (*Cheng and Gartenberg, 2000*), and the ability of increased levels of a transcriptional activator to overcome the silenced state (*Aparicio and Gottschling, 1994*; *Ahmad and Henikoff, 2001*; *Xu et al., 2006*). This model could explain why losses of silencing were more frequent when cells contained only half the normal gene dosage of *SIR1*, *SIR3* or *SIR4*. Perhaps losses of silencing at *HML* and *HMR* resulted from stochastic fluctuations in the local concentration of silencing factors. Alternatively, specific perturbations such as DNA replication may have disrupted the binding of Sir proteins to nucleosomes, thereby creating an opportunity for RNA polymerase to access heterochromatin.

## Defining the roles of Sir1 in silencing

The contribution of Sir1 to silencing in wild-type cells was previously underappreciated. Heritability of the repressed state in a subset of *sir1Δ* cells showed that Sir1 was not absolutely required for the maintenance or inheritance of silenced chromatin, and suggested that Sir1 primarily functioned in establishment (*Pillus and Rine, 1989*). However, the presumed role of Sir1 seemed paradoxical under the assumption that wild-type cells never lose silencing and thus do not undergo events requiring re-establishment. This point of confusion can now be resolved by two key findings. First, the Cre-based assay revealed that transient transcription of *HML* and *HMR* occurred at a low rate in wild-type cells. Therefore, Sir1 may serve a purpose in the recovery of silencing on the rare occasion that it is lost. Secondly, losses of silencing occurred more often in cells lacking Sir1, indicating that Sir1 contributed to either the maintenance or inheritance of silencing, or both.

Previous measurements of silencing established that two phenotypic states exist within a population of *sir1* cells (*Pillus and Rine, 1989*; *Xu et al., 2006*), yet the levels of expression in each of the two subpopulations remained unresolved. For example, in *sir1Δ* cells containing a fluorescent reporter under control of the *URA3* promoter at *HML*, the fluorescence intensity profile of each subpopulation shifted in relation to the profiles of wild-type cells and *sir3Δ* cells depending on whether the *URA3*

trans-activator Ppr1 was present (*Xu et al., 2006*). By directly measuring the transcription of a gene from the native *α2* promoter at *HML* with single-cell and single-molecule resolution, we showed that the silenced subset of *sir1Δ* cells was fully repressed, whereas the expressed subset of *sir1Δ* cells transcribed *cre* to the level of *sir4Δ* cells.

Together, the Cre-based assay and the RNA FISH measurements resolved two separable aspects of silencing: the level of repression and the rate at which it is lost. Deletion of *SIR1* affected the rate of silencing loss, but not the level of repression since approximately half of all *sir1* cells completely lacked *cre* transcripts. Moreover, the persistence of RFP expression within regions of *sir1Δ* colonies containing either *HMLα::cre* or *HMRα::cre* and the RFP-GFP cassette underscored the remarkable ability of heterochromatin to template its own replication for multiple cell divisions.

## Is there a role for a dynamic dimension to heterochromatic gene silencing?

In principle, the dynamic nature of silenced chromatin could result from stochastic processes intrinsic to, for example, the binding constants of Sir proteins for their nucleosomes and serve no useful role. Alternatively, the stability of silencing at *HML* and *HMR* could be tuned high enough to achieve cell-type specificity, yet just low enough to allow for events such as DNA replication or mating-type interconversion. Haploid cells of a homothallic strain can switch mating types through repair of a double-stranded DNA break at *MAT* using *HML* or *HMR* as a template. Therefore, successful switching requires that the invading *MAT* strand, along with the accessory proteins involved in recombinational repair, gain access to the heterochromatic donor sequence. We speculate that the dynamics of silenced chromatin may contribute to the efficiency of mating-type interconversion.

## Role of the Hst3 sirtuin and its histone H3 target in heterochromatin stability

The stability of silencing at *HML* depended in part on Hst3 and the deacetylation of H3 K56-ac. H3 K56 localizes to the DNA entry/exit region of the nucleosome (*Luger et al., 1997*), and in the acetylated state it is thought to promote transient DNA 'breathing', during which the DNA partially unwraps from the nucleosome (*Neumann et al., 2009*; *Simon et al., 2011*; *North et al., 2012*). Furthermore, H3 K56-ac is important for transcriptional activation (*Williams et al., 2008*; *Värv et al., 2010*), and telomeric heterochromatin is more accessible to the bacterial *dam* DNA methylase in various H3 K56 mutants (*Xu et al., 2007*). Collectively, these observations suggest that H3 K56-ac destabilizes the nucleosome conformation and thereby renders the DNA more accessible to RNA polymerase.

H3 K56 acetylation and Hst3 expression are both regulated by the cell cycle. Whereas H3 K56 acetylation peaks during S phase (*Masumoto et al., 2005*), Hst3 expression does not peak until $G_2$/M phase (*Spellman et al., 1998*; *Celic et al., 2006*; *Maas et al., 2006*). Therefore, heterochromatin may be especially susceptible to transcription during the time after K56-acetylated histone H3 is deposited into chromatin, but before Hst3 removes the modification.

## A tractable tool for measuring the dynamics of heterochromatic repression

As previous measurements of *HML* silencing were unable to distinguish *hst3* mutants from wild type, our detection of a stability phenotype in this mutant underscored the importance of measuring fluctuations in repression for a fuller understanding of heterochromatin. The ability of this assay to capture the existence of transient events through their conversion to permanent, heritable marks can now allow a comprehensive evaluation of the contribution of any and all genes to heterochromatin dynamics. By preserving a historical record of transcription, the Cre-based assay also provides the opportunity to measure the effects of transient environmental or cellular stresses on silencing subsequent to their occurrence. Such analyses will inform our understanding of the mechanisms by which cells maintain integrity of the epigenome.

Measuring the frequency of half-sectored colonies allowed for quantification of the rate at which silencing was lost. In general, half-sector frequencies were consistent with the overall patterns of colony sectoring. However, strains bearing the histone H3 K56R substitution, for example, were indistinguishable from wild type by sectoring pattern, yet showed a significant increase in the frequency of half-sectored colonies. While we do not fully understand why silencing in histone H3 K56R mutants appeared to be less stable at the two-cell stage of colony growth than at later stages, the complexities of colony development may offer a simple explanation. The physiology of cells during the first cell division of colony

growth, at which point silencing was measured by half-sector analysis, would be expected to differ from the physiology of cells during later divisions. Therefore, it is possible that the histone H3 K56R mutation rendered heterochromatin more sensitive to the physiological state of cells. Overall, both assays were informative, and sectoring patterns in particular provided the opportunity to survey heterochromatin dynamics throughout the various microenvironments and metabolic states of colonies.

A direct comparison could not be made between the rate of RFP-to-GFP switching, as determined by half-sector analysis, and the frequency of wild-type cells expressing *HMLα::cre*, as determined by FISH. Whereas half-sector analysis measured events throughout the duration of an entire cell cycle, the FISH method measured expression in a snapshot of time. Furthermore, the background levels of FISH signal in the negative control strain, though very low, limited quantitative interpretation of the low *cre* RNA levels in wild-type cells. The FISH measurements did show, however, that any detectable transcription of *HMLα::cre* in wild-type cells was both rare and restricted to relatively low levels. These observations implied that a small number of *cre* transcripts produced enough recombinase to catalyze the RFP-to-GFP switch. Further experiments would be necessary to determine the number of *cre* RNAs sufficient for recombination.

## Capturing losses of silencing in real time

The live-cell imaging of pedigrees of wild-type cells endowed with the Cre-based assay reflected our first efforts to visualize in real time the silencing loss events whose history is written in the number and size of green colony sectors. In principle, RNA FISH by itself could provide better time resolution, but morphological classification of fixed spheroplasts using brightfield images was unreliable, especially with the high density of cells needed to capture rare events. Despite the time lag between loss-of-silencing events and the production of the GFP chromophore, several points were clear. First, there was no obvious mother-daughter bias as to which cell contained the more labile *HML* locus. Secondly, the expression of *cre* and the subsequent switches from red to green could occur within the same cell cycle. Overall, the variety of patterns we observed were consistent with the possibility that losses of silencing occur at multiple points during the cell cycle, but such conclusions rely on assumptions for which we have no independent verification. Knowing the extent of variation in the time lag between *HMLα::cre* expression and recombination would help solidify further interpretation. Intriguingly, there were three pedigrees in which a red daughter, born from a mother that switched to green, itself switched to green. Given the low probability of a switch, two switches in such closely related cells implied there was either perdurance of the Cre protein following a single loss-of-silencing event, or some 'heritable' instability shared between these two cells, as has been previously suggested in a different context (*Kaufmann et al., 2007*). Future experiments such as live-cell imaging of transcription should reveal the exact timing of silencing loss and address whether the history of a lineage affects stability.

# Materials and methods

## Yeast strains

The strains, plasmids, and oligonucleotides used in this study are listed in *Supplementary files 1, 2 and 3*, respectively. All strains were derived from the W303 background. Deletions were made using one-step integration of gene disruption cassettes (*Longtine et al., 1998*; *Goldstein and McCusker, 1999*; *Gueldener et al., 2002*) and confirmed by PCR of the 5′ and 3′ junctions. For analysis of *HHT2* (histone H3) mutations, the *URA3*-marked plasmid (pJR2657) in JRY9638 was replaced with a *TRP1*-marked plasmid (either pJR2759, pJR3212 or pJR3213) by plasmid shuffle to produce JRY9639, JRY9640 and JRY9641. Using the *hst3Δ::K.l.LEU2* fwd/rev primers and pUG73 (*Gueldener et al., 2002*), *HST3* was deleted in each of these strains to create JRY9736, JRY9737 and JRY9738.

The *cre* gene from bacteriophage P1 was integrated at *HML* by transformation of a *sir4Δ* strain containing *HMLα::K.l.URA3* and counter-selection on medium containing 5-fluoroorotic acid (5-FOA). First, the *K.l.URA3* gene was amplified by PCR from pUG72 (*Gueldener et al., 2002*) using the *HMLα::K.l.URA3* fwd/rev primers and integrated at *HMLα*, replacing the coding sequence of the *α2* gene. Then, the *cre* gene was amplified from pSH47 (*Güldener et al., 1996*) using the *HMLα::cre* fwd/rev primers, and the resulting PCR product was transformed into the strain containing *HMLα::K.l.URA3*. 5-FOA was used to select for *HMLα::cre* cells, which no longer contained *K.l.URA3*. Integration of *cre* was confirmed by sequencing. The resulting strain was mated with a *SIR4* strain and sporulated to obtain a haploid containing both *SIR4* and the *HMLα::cre* allele.

To construct the *HMRα::cre* allele, the *K.l.URA3* gene was amplified by PCR from pUG72 (*Gueldener et al., 2002*) using the *HMRa::K.l.URA3* fwd/rev primers and integrated at *HMRa* in a *sir4Δ* strain, thus replacing the coding sequence of the *a1* gene with *K.l.URA3*. Then, the *HMRα::cre* fwd/rev primers were used to amplify a fragment of *HMLα::cre* from JRY9628 genomic DNA that spanned from the X region to the Z1 region (see *Figure 1A*), and the resulting PCR product was transformed into the strain containing *HMRa::K.l.URA3*. 5-FOA was used to select for *HMRα::cre* cells, which no longer contained *K.l.URA3*. Integration of *cre* was confirmed by sequencing. The resulting strain was mated with a *SIR4* strain and sporulated to obtain a haploid containing both *SIR4* and the *HMRα::cre* allele.

For integration of the RFP-GFP cassette at the *URA3* locus, the $P_{GPD}$-*loxP*-*yEmRFP*-$T_{CYC1}$-*kanMX*-*loxP*-*yEGFP*-$T_{ADH1}$ sequence in pJR3214 was amplified by PCR using the *ura3Δ::RFP-GFP* fwd/rev primers. pJR3214 contains the following features: the *ampR*-containing fragment of pUG73 (*Gueldener et al., 2002*) resulting from digestion with restriction enzymes NdeI and NotI (2301 bp); the *GPD* promoter from p413-GPD (*Mumberg et al., 1995*); a *loxP* site (5'-ATAACTTCGTATAGCATA CATTATACGAAGTTAT-3'), separated from $P_{GPD}$ by a SalI restriction site (GTCGAC); immediately down-stream of the first *loxP* site is the *yEmRFP* gene (*Keppler-Ross et al., 2008*), followed by a BglII restriction site (AGATCT) and the *CYC1* terminator from pSH63 (*Gueldener et al., 2002*); the *kanMX* cassette from pUG6 (*Güldener et al., 1996*), separated from $T_{CYC1}$ by a StuI restriction site (AGGCCT); adjacent to the *kanMX* cassette is a second *loxP* site (5'-ATAACTTCGTATAGCATACATTATACGAAGTTAT-3'); and immediately following the second *loxP* site is an EcoRV restriction site (GATATC) and the *yEGFP*-$T_{ADH1}$ sequence from pKT127 (*Sheff and Thorn, 2004*).

The version of the RFP-GFP cassette marked with *hphMX* instead of *kanMX* (in strains JRY9729 and JRY9730) was constructed by transforming strains containing the *kanMX*-marked RFP-GFP cassette with PCR product amplified from pAG32 (*Goldstein and McCusker, 1999*) using the $P_{TEF}$ fwd and $T_{TEF}$ rev primers.

## Colony growth and imaging

All colonies were imaged using a Zeiss Axio Zoom.V16 microscope equipped with ZEN software (Zeiss, Jena, Germany), a Zeiss AxioCam MRm camera and a PlanApo Z 0.5× objective. The colonies shown in *Figures 2, 6 and 7* (and the associated figure supplements) were imaged on day 6 of growth, whereas the colonies shown in *Figures 3 and 5* (and the associated figure supplements) were imaged on day 3 of growth. Images were assembled using Photoshop (Adobe Systems, San Jose, CA).

In preparation for colony analysis, red cells were selected using medium containing G418 (Geneticin) (Life technologies, Grand Island, NY). Due to the complete loss of silencing in *sir2Δ* cells, JRY9633 was unable to grow in the presence of G418 and was therefore grown on nonselective medium. Cells were then grown to log phase in Complete Supplement Mixture (CSM) −Trp (Sunrise Science Products, San Diego, CA), and serial dilutions were performed to spread approximately 10 cells/plate (CSM −Trp, 1% agar). To inhibit sirtuin activity, 200 µl of 0.5 M nicotinamide was spread onto CSM −Trp, 1% agar plates prior to plating cells.

For *Figure 5* and *Figure 5—figure supplement 1*, red cells of JRY9729 and JRY9730 were selected using medium containing both G418 (Life technologies) and Hygromycin B (Sigma–Aldrich, St. Louis, MO), and then sporulated for 4–5 days at room temperature on 1% potassium acetate. The resulting tetrads were dissected on CSM −Trp plates and grown into colonies for imaging.

For half-sector analysis, red cells were selected using G418 and then grown to log phase in CSM −Trp. Serial dilutions were performed to plate approximately 100 cells/plate (CSM −Trp). On day 3 of growth, plates were scanned, face up, with a Typhoon Trio (GE Healthcare Life Sciences, Buckinghamshire, England) using the 488-nm laser and 520-nm emission filter to detect GFP fluorescence. Since colonies of cells containing the H3 K56R substitution grew notably slower than colonies of other genotypes, plates were scanned on day 4 of growth for JRY9640 and JRY9737. Colonies that were half green or completely green were manually counted. The total number of colonies was determined with Matlab (MathWorks, Natick, MA) by applying a threshold and converting scans to a binary image (code available upon request). Then, the following equation was used to calculate the half-sector frequency:

# of half-sectored colonies/(# of total colonies–# of fully green colonies)

Three independent experiments were performed for each genotype. See *Table 1—source data 1* for the total number of colonies, as well as the number of half-sectored colonies and fully green colonies determined for every experiment.

## Flow cytometry

Colonies were grown as described above, scraped off the agar surface and resuspended in synthetic complete (SC) medium (Sunrise Science Products). After growth overnight, cultures were diluted back to 0.01 $OD_{600}$ and then grown in SC to approximately 0.2 $OD_{600}$. Cells were harvested by centrifugation, and then washed and resuspended in PBS, pH 7.4 on ice. A FC-500 flow cytometer (Beckman–Coulter, Brea, CA) was used to measure the GFP fluorescence intensity of $10^5$ cells/sample. Using FlowJo software (Tree Star, Ashland, OR) cells were gated based on forward and side scatter, and the data were exported as FCS files. Data analysis was performed with Matlab (Mathworks).

## RNA preparation for quantitative RT-PCR

Total RNA was isolated from log-phase cells using hot acidic phenol (*Collart and Oliviero, 2001*). RNA was digested with DNase I (Roche Diagnostics, Indianapolis, IN) and then purified by phenol-chloroform extraction and precipitation with isopropanol. cDNA was synthesized using the SuperScript III First-Strand Synthesis System for RT-PCR (Invitrogen, Carlsbad, CA) and oligo(dT) primers. Quantitative PCR of cDNA was performed using the Thermo Scientific DyNAmo HS SYBR Green qPCR Kit (Fisher Scientific, Chicago, IL) and a Mx3000P machine (Stratagene, acquired by Agilent Technologies, Santa Clara, CA). Samples were analyzed in technical triplicate for three independent RNA preparations.

## Single-molecule RNA FISH

RNA FISH was performed as described (*Raj et al., 2008*) using JRY9630, JRY9631, JRY9632 and JRY4012. All probes (see *Supplementary file 4* for probe sequences) were designed, synthesized and labeled by Stellaris (Biosearch Technologies, Novato, CA). Probes targeting *cre* were coupled to Quasar 670 dye (Biosearch Technologies) and probes targeting *KAP104* were coupled to CAL Fluor Red 590 dye (Biosearch Technologies). Each probe mix (5 nmol) was dissolved in 100 µl of TE buffer, pH 8.0. For hybridization, 1:10 dilutions of the *cre* and *KAP104* probe stocks were made, and 1 µl of each probe dilution was added to 50 µl of hybridization buffer.

Cells were grown in CSM to 0.2–0.3 $OD_{600}$ and then fixed and hybridized using the protocol as described (*Youk et al., 2010*) with additional guidance (*Raj, 2013*) and the following modification to the spheroplasting procedure: cells were digested with 3 µl of zymolyase (2.5 mg/ml) at 30°C and monitored by phase contrast microscopy until approximately 80% of cells appeared to be digested (35–50 min) (L Teytelman, personal communication, January 2013). The *cre* and *KAP104* probes were hybridized to cells overnight at 30°C in 10% formamide buffer. Samples were then washed, stained with DAPI, washed again and resuspended in 10 µl of glucose-oxygen-scavenging (GLOX) solution without enzymes. Prior to imaging, 10 µl of GLOX solution with enzymes (1% [vol/vol] catalase [Sigma-Aldrich C3515], 1% [vol/vol] glucose oxidase [Sigma-Aldrich G2133], 2 mM Trolox [Sigma-Aldrich 238813]) was added to the sample.

Images were acquired with a DeltaVision RT microscope and softWoRx software (Applied Precision, Issaquah, WA) using a 60 × /1.40 oil-immersion objective (Olympus, Tokyo, Japan), a CoolSNAP HQ CCD camera (Photometrics, Tuscon, AZ) and the following filters from Chroma Technology (Bellows Falls, VT): ET402/15x, ET455/50m (DAPI); ET555/25x, ET605/52m (CAL Fluor Red 590); and ET645/30x, ET705/72m (Quasar 670). Series of z-stacks were acquired with a step size of 0.2 µm. Cell boundaries were hand-drawn using ImageJ software (NIH, Bethesda, MD). Spots were detected and analyzed using Matlab (MathWorks). First, a three-dimensional Laplacian of Gaussian filter (*Raj et al., 2008*) was applied to raw images of *cre* and *KAP104* RNA. Then, a fluorescence intensity threshold was selected to identify spots within an expected range of sizes. Given that some *sir4Δ* and *sir1Δ* cells contained a relatively large number of *cre* transcripts, some of which appeared to overlap each other in a maximum-intensity projection of z-stacks, spot detection was performed in three dimensions to resolve individual spots (code adapted from L Teytelman and available upon request). Within each experiment (three independent experiments were performed), the same threshold values (one for *cre* RNA images and one for *KAP104* RNA images) were applied across all samples.

## Live-cell imaging

Cells containing *HMLα::cre* and the RFP-GFP cassette (JRY9628) were grown to mid-log phase in CSM −Trp, sonicated once for 5 s at a power output of 10%, and then imaged over time using the CellASIC ONIX Microfluidic Platform (EMD Millipore, Hayward, CA). Growth was restricted to a single focal plane within chambers of the Y04C Microfluidic Plate for Haploid Yeast (EMD Millipore). CSM −Trp

flowed through the chambers continuously at a pressure of 3 psi. Brightfield and fluorescence images of 46 different fields were taken every 10 min for 13 hr using MetaMorph software (Molecular Devices, Sunnyvale, CA) and an Eclipse Ti microscope (Nikon Instruments, Melville, NY) equipped with a Clara Interline CCD camera (Andor Technology, Belfast, Northern Ireland) and a CFI Apo TIRF 60 × /1.49 oil-immersion objective (Nikon Instruments), which was heated to 30°C using an objective heater (Bioptechs, Butler, PA). Exposure times for yEGFP and yEmRFP fluorescence were each 100 ms. Image analysis and video assembly were performed using ImageJ software (NIH).

## Acknowledgements

We thank Leonid Teytelman, Alexander van Oudenaarden and Abby Dernburg for their valuable guidance and provisions for the RNA FISH. We also thank Jack Wang for strain contributions, Oskar Hallatschek for use of the Zeiss Axio Zoom.V16, Liam Holt for use of the microfluidic platform and the Berkeley Stem Cell Center for use of the flow cytometer. We are grateful to members of the Rine laboratory for helpful discussions, especially Meru Sadhu, David Steakley, Sarah Bissonnette, Deborah Thurtle and Gavin Schlissel. We thank Doug Koshland and Georjana Barnes for critical review of the manuscript.

## Additional information

### Funding

| Funder | Grant reference number | Author |
| --- | --- | --- |
| National Institutes of Health | GM 031105 | Jasper Rine |
| National Institutes of Health | 5 T32 GM 7232-34 | Anne E Dodson |
| National Science Foundation | DGE 1106400 | Anne E Dodson |

The funders had no role in study design, data collection and interpretation, or the decision to submit the work for publication.

### Author contributions

AED, Conception and design, Acquisition of data, Analysis and interpretation of data, Drafting or revising the article; JR, Conception and design, Analysis and interpretation of data, Drafting or revising the article

### Author ORCIDs

Anne E Dodson, http://orcid.org/0000-0001-7719-0496

## Additional files

### Supplementary files

• Supplementary file 1. Table of yeast strains used in this study. Strain JRY4012 originated from *Fox et al. (1995)*. All other strains were constructed for this study.

• Supplementary file 2. Table of plasmids used in this study. Plasmid pJR3214 was constructed for this study. Plasmid pJR2657 originated from *Dion et al. (2005)*. All other plasmids originated from the Rine laboratory plasmid collection.

• Supplementary file 3. Table of oligonucleotides used in this study.

• Supplementary file 4. Table of FISH probes used in this study.

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
