## [Decision Letter]

Thank you for sending your work entitled “Heritable capture of heterochromatin dynamics in *Saccharomyces cerevisiae*” for consideration at *eLife*. Your article has been favorably evaluated by Richard Losick (Senior editor), Dan Gottschling (guest Reviewing editor), and 2 reviewers.

The Reviewing editor and the reviewers discussed their comments before we reached this decision, and the Reviewing editor has assembled the following comments to help you prepare a revised submission.

This manuscript presents extremely interesting results concerning the regulation of transcription of the silent mating type cassettes in *S. cerevisiae*. By using the Cre–regulated RFP–GFP reporter described in the paper, the authors can stably capture exceedingly rare expression from within a silent mating type cassette. Using this tool, they investigate the frequency of expression of these cassettes in both wild–type cells and in a set of mutants. One highlight of the manuscript is that the system seems to work and work well. It shows that there is a very low level of expression in wild–type that was not previously detectable. A second highlight is that the experiments reveal that Sir1 is required for more than establishment of silencing, as the data reveal that it is also likely required for maintenance. Other results provide evidence that the histone deacetylase Hst3 is required for silencing and that the absolute levels of heterochromatin components—histones and Sir proteins—are also important. Overall, the development of the reporter system as well as the specific results will be of interest to a large number of scientists.

The referees agreed that the data are interesting and overall of good technical quality. However, there was also a consensus that the promise of quantitative measurement was not carried forward in the mutant analysis, and that this should be done before acceptance of the manuscript. This and a few other concerns to be addressed are presented below:

1) Many of the mutant phenotypes in the manuscript are not described in a very quantitative fashion, as they are described and shown only as increased levels of green sectoring. This is true for effects of Sir protein levels, H3/H4 levels, Hst3 effects, H3K56Q mutants, and more. But one of the important points of this manuscript is the quantitative nature of the assay. The reviewers see this initial work as *the* chance to set the standard in the field for all other follow up studies using this assay. Thus before endorsing this work for acceptance, all the mutant analyses should be carried out via a half sector analysis to provide a quantitative measurement for comparison (rather than the semi–quantitative analysis of relative sectoring that was presented). This will help alleviate uninterpretable comparisons that occur between labs or even between mutants.

2) It is clear that the expression of GFP is dependent upon Cre expression. However, the level of Cre expression and the efficiency at which it mediates recombination at the lox sites is not clear. That is, the FISH experiments tell us that very few wt cells have a Cre signal above background (2–4 spots versus 1 spot). However they do not tell us whether 2–4 spots of Cre are sufficient for recombination. At best this can only be inferred from comparing the frequency of FISH signals to half–sectors. Please acknowledge/define these limitations of your interpretation in the text.

3) There are a few concerns/comments about the Hst3/H3K56 experiments. First, in Figure 10, the authors conclude that an increased level of sectoring is caused by reduced levels of H3/H4 based on comparing wild–type strains to those with only one copy of H3/H4 on a plasmid. A change in H3/H4 level seems like the likeliest explanation, but other explanations seem possible in the absence of careful measurements of histone levels. Second, the analysis of the amino acid changes at H3K56 (Figure 9) appears to have been done in strains with this same configuration of H3/H4 genes; that is, with a perturbation of sectoring levels already in place. It's hard to know if the altered histone gene configuration might affect the degree of the change caused by the H3K56 changes. It is suggested these two sections be combined to present the results and possible caveats more clearly. Ideally, what is currently Figures 8–10 would be part of the same figure. Finally, there's evidence that an Hst4 deletion causes little phenotype except when combined with an Hst3 deletion, in which case it enhances the Hst3 phenotype. Did the authors test an Hst3 Hst4 double mutant for sectoring levels? If they did not, it would still be helpful to acknowledge the possibility that the double mutant might show an even stronger phenotype.

---

## [Author Response]

*1) Many of the mutant phenotypes in the manuscript are not described in a very quantitative fashion, as they are described and shown only as increased levels of green sectoring. This is true for effects of Sir protein levels, H3/H4 levels, Hst3 effects, H3K56Q mutants, and more. But one of the important points of this manuscript is the quantitative nature of the assay. The reviewers see this initial work as* the *chance to set the standard in the field for all other follow up studies using this assay. Thus before endorsing this work for acceptance, all the mutant analyses should be carried out via a half sector analysis to provide a quantitative measurement for comparison (rather than the semi–quantitative analysis of relative sectoring that was presented). This will help alleviate uninterpretable comparisons that occur between labs or even between mutants*.

Our top priority was to quantify the rate of silencing loss for all mutants by measuring the frequency of half-sectored colonies. Measuring these low rates was a daunting and laborious task that in the end required hundreds of liters of media and the screening of hundreds of thousands of colonies. Hence it took a substantial amount of time to revise the manuscript. In anticipation of this workload, we excluded the few mutants that did not show a sectoring phenotype (the *sir2* hemizygote and the *hst1Δ*, *hst2Δ* and *hst4Δ* mutants) from our analysis. The newly determined half-sector frequencies are represented as bar graphs in new panels of Figures 5, 6 and 7. A summary of these values is presented in Table 1, and the raw data are available in [Supplementary-material SD1-data].

For the majority of measurements, the results were straightforward. Differences in sectoring patterns were reflected by differences in the frequency of half-sectored colonies for the genotypes shown in Figures 3, 5, 6 and 7. For Figure 7, however, we found that some genotypes (in particular, strains containing the histone H3 K56R substitution) displayed frequencies of half-sectored colonies that were unexpected based upon the sectoring phenotypes, thus challenging us to consider the ways in which half-sector occurrence could have diverged from the overall sectoring pattern. The most parsimonious explanation for these results was, given that cells experience distinct physiological states between different stages of colony growth and even between different locations within the same colony, these changes in state may have affected silencing in certain genetic backgrounds. Whereas half-sector measurements were restricted to the first cell division of colony growth, the sectoring patterns of developed colonies offered a more complex yet expansive interpretation of heterochromatin dynamics. We added this point to the Discussion.

*2) It is clear that the expression of GFP is dependent upon Cre expression. However, the level of Cre expression and the efficiency at which it mediates recombination at the lox sites is not clear. That is, the FISH experiments tell us that very few wt cells have a Cre signal above background (2–4 spots versus 1 spot). However they do not tell us whether 2–4 spots of Cre are sufficient for recombination. At best this can only be inferred from comparing the frequency of FISH signals to half–sectors. Please acknowledge/define these limitations of your interpretation in the text*.

FISH measurements showed that in the rare wild-type cells that transcribed *HMLα::cre* , the level of *cre* RNA ranged from 1 to 4 molecules per cell, which was low in comparison to the average level of *cre* RNA in the *sir4Δ* mutant. We inferred from this observation that very few *cre* transcripts were sufficient to induce a RFP-to-GFP switch. Although we stand by this interpretation, we agree with the reviewers that more evidence is needed to confirm this possibility. Therefore, we moved this text from the Results to the Discussion and elaborated on the limitations of comparing the FISH analysis to the Cre-based assay.

*3) There are a few concerns/comments about the Hst3/H3K56 experiments. First, in Figure 10, the authors conclude that an increased level of sectoring is caused by reduced levels of H3/H4 based on comparing wild–type strains to those with only one copy of H3/H4 on a plasmid. A change in H3/H4 level seems like the likeliest explanation, but other explanations seem possible in the absence of careful measurements of histone levels. Second, the analysis of the amino acid changes at H3K56 (Figure 9) appears to have been done in strains with this same configuration of H3/H4 genes; that is, with a perturbation of sectoring levels already in place. It's hard to know if the altered histone gene configuration might affect the degree of the change caused by the H3K56 changes. It is suggested these two sections be combined to present the results and possible caveats more clearly. Ideally, what is currently Figures 8–10 would be part of the same figure. Finally, there's evidence that an Hst4 deletion causes little phenotype except when combined with an Hst3 deletion, in which case it enhances the Hst3 phenotype. Did the authors test an Hst3 Hst4 double mutant for sectoring levels? If they did not, it would still be helpful to acknowledge the possibility that the double mutant might show an even stronger phenotype*.

For the experiments related to Hst3 and histone H3 K56, we combined what were previously Figures 8 and 9 into what is now Figure 7 to show the sirtuin deletion mutants together with the histone H3 K56 mutants, as suggested. Figure 10, which showed that a strain containing only one copy of the histone H3-H4 gene pair on a plasmid exhibited more sectoring than a strain containing the two endogenous histone H3-H4 gene pairs, was turned into a supplementary figure to accompany the new Figure 7. We added text to both the legend for Figure 7 and the main text to clarify that all strains shown in Figure 7 were in the genetic background containing only one copy of the histone H3-H4 gene pair on a plasmid. This background did sensitize the sectoring phenotype of *hst3Δ* colonies as it did for wild-type colonies (compare the *hst3Δ* colonies between panels A and B in Figure 7—figure supplement 1), but it had a more variable effect on the frequency of half sectors, which is now discussed in the Results. A change in histone H3 and H4 levels is indeed the likeliest explanation for the histone H3-H4 gene dosage effect on silencing. However, to address the valid point that alternative explanations cannot be ruled out, we adjusted the text to both strengthen support for this model and recognize any inherent assumptions. Finally, since previous publications have shown Hst4 to be a minor contributor to the role of its paralog Hst3, we agree with the reviewers that deletion of Hst4 would likely enhance the phenotype of the *hst3Δ* mutant. In recognition of this, we added text to the Results.